# Introducing Polar Groups in Porous Aromatic Framework for Achieving High Capacity of Organic Molecules and Enhanced Self-Cleaning Applications

**DOI:** 10.3390/molecules27186113

**Published:** 2022-09-19

**Authors:** Zhuojun Yan, Yimin Qiao, Qiqi Sun, Bo Cui, Bin Feng, Naishun Bu, Kuo Chu, Xianghui Ruan, Ye Yuan, Yajie Yang, Lixin Xia

**Affiliations:** 1College of Chemistry, Liaoning University, Shenyang 110036, China; 2School of Environmental Science, Liaoning University, Shenyang 110036, China; 3Key Laboratory of Polyoxometalate and Reticular Material Chemistry of Ministry of Education, Northeast Normal University, Changchun 130024, China; 4Yingkou Institute of Technology, Yingkou 115014, China

**Keywords:** superhydrophobic, Sonogashira–Hagihara coupling, porous aromatic framework, oil/water separation, organic pollution

## Abstract

Due to the frequent oil/organic solvent leakage, efficient oil/water separation has attracted extensive concern. However, conventional porous materials possess nonpolar building units, which reveal relatively weak affinity for polar organic molecules. Here, two different polarities of superhydrophobic porous aromatic frameworks (PAFs) were synthesized with respective orthoposition and paraposition C=O groups in the PAF linkers. The conjugated structure formed by a large number of alkynyl and benzene ring structures enabled porous and superhydrophobic quality of PAFs. After the successful preparation of the PAF solids, PAF powders were coated on polyester fabrics by a simple dip-coating method, which endowed the resulting polyester fabrics with superhydrophobicity, porosity, and excellent stability. Based on the unique structure, the oil/water separation efficiency of two superhydrophobic flexible fabrics was more than 90% for various organic solvents. Polar LNU-26 PAF showed better separation performance for the polar oils. This work takes the lead in adopting the polar groups as building units for the preparation of porous networks, which has great guiding significance for the construction of advanced oil/water separation materials.

## 1. Introduction

With the rapid development of industry and transport, frequent organic waste and oil leakage has caused serious panic [1,2,3,4]. Oil spills will cause water pollution and affect marine ecology, harming the health of living bodies. Therefore, there is an urgent need to develop an effective system to deal with the above problems for environmental remediation. The adsorption method is an effective method that is currently widely used to deal with organic pollution, because of certain advantages, including easy availability of adsorbent materials and simple practical operation. So far, many commercially available porous materials are prepared, such as activated carbon [5], zeolite [6], inorganic–organic hybrid solid [7], and carbon nanotube [8,9]. However, these materials have some drawbacks of limited absorption capacity and poor affinity.

Porous aromatic frameworks (PAFs) are well known because of their easily tunable pore structure, high stability, and large specific surface area. Various PAFs with different structures and functions have been designed and synthesized as porous media for various applications, such as gas separation [10], supercapacitor [11], and catalysis [12]. Notably, most of these solids are superhydrophobic, referring to a solid with a water contact angle (WCA) >150°, which is an important characteristic in meeting oil/water separation [13,14,15,16,17,18,19,20,21,22,23]. However, most of the reported PAF samples are generally constructed by nonpolar aromatic rings as building units. These structural units possess relatively weak affinity for polar organic molecules, leading to a suppressed adsorption and removal capability.

In this contribution, we synthesized two different polarities of superhydrophobic PAF solid powders with respective orthoposition and paraposition C=O groups in the PAF linkers. The obtained PAF powders were coated on polyester fabrics through a simple dip-coating method to prepare two superhydrophobic fabrics. The separation efficiency of the prepared PAF-coated materials for aromatic organic molecules is over 90%, which shows great potential in oil/water separation or pollutant removal applications.

## 2. Results

Two different polarities of PAF samples were synthesized through Sonogashira–Hagihara cross-coupling reaction of 1,3,5-triethynylbenzene and heterocyclic isomers bearing orthoposition and paraposition C=O groups, denoted as LNU-26 and LNU-27, respectively (Figure 1). The chemical structures of both LNU-26 and LNU-27 were confirmed by Fourier-transform infrared (FT-IR) and ^13^C solid-state NMR spectroscopy. The disappeared characteristic signals at ~460 cm^−1^ (C–Br stretching band) and 3300 cm^−1^ (C≡C–H stretching band) together with the appearance of the –C≡C– stretching vibration at ~2200 cm^−1^ demonstrated the successful polymerization of the Sonogashira reaction (Figure 1a,b). As illustrated in ^13^C solid-state NMR spectra, the resonances in the range of 120–150 ppm indicated the substituted and unsubstituted carbon atoms on aromatic rings; the chemical shifts at 80–100 ppm were attributed to –C≡C– groups in the PAF networks. In the meantime, the existence of C=O units was observed at ~180 ppm (Figure 1c). All these results proved the successful preparation of the Sonogashira–Hagihara reaction and the structural integrity of PAF networks.

The physical and chemical stability of the superhydrophobic samples are two important factors in practical applications. As depicted in Figure 1d, neither LNU-26 nor LNU-27 revealed distinctive XRD peaks, indicating that the PAF backbone was amorphous. PAF solids could not be dissolved or decomposed in various solvents, including methanol, ethanol, acetone, dichloromethane, chloroform, DMF, and tetrahydrofuran, and so on. From the TGA curves, the weight changes of the two LNU samples were not observed before 300 °C, and the residual weights at 950 °C were close to 60% (Appendix A), which suggested ultrahigh chemical and thermal stability of PAF solids. The morphologies of the as-prepared superhydrophobic PAF solids were studied using scanning electron microscopy (SEM). LNU-26 was formed by the accumulation of bulk solids, and LNU-27 was composed of fibrous solids (Appendix A). According to transmission electron microscopy (TEM), both LNU-25 and LNU-26 possessed a wormlike structure (Appendix A). The porosity of the resulting polymers were investigated by N_2_ adsorption–desorption analysis at 77 K; both materials belong to the type II/IV isotherm according to the IUPAC classification (Figure 2a) [24]. The calculated Brunauer–Emmett–Teller (BET) surface areas were found to be 44.8 and 33.9 m^2^ g^−1^ for LNU-26 and LNU-27, respectively. As shown in Figure 2b, the pore size distribution curves of two materials have similar trends according to the NLDFT model, indicating the micro-mesopores of PAF architecture.

To explore the hydrophobic properties of PAF solids, we dispersed the powders in a mixture composed of oil and water. For instance, kerosene and water were added to the sample bottle at a volume ratio of 9:20, and then LNU-26 powder was poured into the bottle. It was found that the PAF solids were uniformly dispersed in the kerosene and stayed above the water interface, indicating the lipophilic and hydrophobic nature of PAF solids (Figure 3a). As depicted in SEM images, the surface of polyester fabric was smooth before being coated with PAF solids. After being coated with the PAF powders, the surface of the polyester fabric became rough, and the cracks between the fibers were filled with polymer materials (Figure 3b–d). This result indicated that the PAF solids were successfully coated on the polyester fabric. It was reported in the literature that the particles exhibited a special microstructure observed by SEM in the entire region, and this structure was conducive to enabling a superhydrophobic surface [25]. The WCA of the PAF solids’ coated fabric was tested to be 155.2° for LNU-26 and 154° for LNU-27 (Figure 3e–g inset), showing the high superhydrophobicity of the PAF-solid-coated fabric [26,27,28].

Figure 3e–g is a comparison diagram of dropping water and chloroform droplets on the original polyester fabric and the superhydrophobic polyester fabric coated with PAF solids, respectively. It can be seen from the figures that water and chloroform droplets are completely absorbed in the original polyester fabric, indicating that the polyester fabric is lipophilic and hydrophilic. After water and chloroform are dropped on the PAF-solid-coated polyester fabrics, the water droplets are almost spherical, while the chloroform droplets are completely absorbed, indicating that the PAF-solid-coated polyester fabrics are oleophilic, and the PAF materials maintain excellent superhydrophobicity after being coated on the fabric.

The direct separation of oil/organic wastewater using superhydrophobic materials has attracted extensive attention due to the high oil/water separation efficiency and selectivity. Seven oils or organics with different viscosities were selected (bromobenzene 2.92 mPa·s, 20 °C, to hexane 0.66 mPa·s, 20 °C) to test the oil/water separation efficiency of PAF-solid-coated fabrics. As seen in Appendix A, the raw polyester fabric has no capability to separate oil and water mixture. On the contrary, the separation efficiency of the two PAF-solid-coated fabrics was above 90% for various oils with different viscosities (Figure 4). The better polar oil separation of LNU-26 PAF materials was attributed to the fact that the polar building units adsorbed the polar oil molecules to form a separation membrane [29,30].

Further, the superhydrophobic PAF solids were coated on a glass plate to explore their self-cleaning performance (Figure 5). Using a soil and chalk dust as a pollutant, the soil/chalk dust was sprinkled on the surface of the glass plate, and then the glass plate was inclined 15° to flow the water droplets. For ease of observation, the water droplets were dyed with methyl blue. As illustrated in Figure 5a,d, the untreated glass plates hold the soil and chalk dust due to the strong affinity between water droplets and glass. As for the PAF-solid-coated glass plates, the soil and chalk dust were taken away by the water droplets (Figure 5b,c,e,f). This phenomenon indicated that the glass plate coated with PAF solids had excellent superhydrophobicity, resulting in a good self-cleaning ability.

## 3. Materials and Methods

### 3.1. Materials

1,3,5-Triethynylbenzene (TCI), 2,7-dibromo-9,10-phenanthrenedione (Energy Chemistry), 2,6-dibromoanthraquinone (Energy Chemistry), triethylamine (Energy Chemistry), anhydrous *N,N*′-dimethylformamide (Sinopharm Chemical Reagent Co., Ltd., Shanghai, China), cuprous iodide (Sigma-Aldrich, Darmstadt, Germany), tetrakis(triphenylphosphine)palladium (Sigma-Aldrich, Germany), and all other materials were obtained from commercial suppliers and used without further purification.

### 3.2. Characterization

Fourier-transform infrared spectroscopy (FTIR) was performed using KBr pellets on a Shimadzu Prestige 21 Fourier-transform infrared spectrometer. Solid-state ^13^C-NMR spectrum was measured on a Bruker Avance III 400 WB spectrometer at a MAS rate of 5 kHz. Thermogravimetric analysis (TGA) was tested using a Mettler Toledo TGA/DSC 2 thermal analyzer under nitrogen atmosphere. Powder X-ray diffractometer (PXRD) measurement was carried out on a Bruker D8 Quest diffractometer with Cu-Kα radiation. Scanning electron microscopy (SEM) analysis was conducted on an SU8010 model scanning electron microscope with an accelerating voltage of 5 kV. Transmission electron microscopy (TEM) was recorded on a JEM-2100 with an accelerating voltage of 200 kV. N_2_ adsorption isotherm was obtained on a Micromeritics ASAP 2460 instrument. Contact angle was measured by a contact angle meter (Krüss GmbH DSA1005, Hamburg, Germany).

### 3.3. Synthesis of PAF Solids

PAF solids were synthesized via the Sonogashira–Hagihara coupling reaction (Figure 1). For the LNU-26 sample, 1,3,5-triethynylbenzene (151 mg, 0.998 mmol), 2,7-dibromo-9,10-phenanthrenedione (549 mg, 1.498 mmol), cuprous iodide (10 mg), and tetrakis(triphenylphosphine)palladium (30 mg, 0.026 mmol) were dissolved in a mixture of triethylamine (8 mL) and *N*,*N′-*dimethylformamide (20 mL). Degassed by three freeze–pump–thaw cycles, the mixture was heated to 80 °C for 72 h. After being cooled to room temperature, the mixture was washed with respective chloroform, tetrahydrofuran, ethanol, and acetone several times to remove the unreacted monomers and catalyst residues. Further purification was performed by Soxhlet extraction (tetrahydrofuran, chloroform, and dichloromethane) for 72 h, followed by drying at 90 °C for 24 h. The LNU-27 sample was obtained by replacing 2,7-dibromo-9,10-phenanthrenedione with 2,6-dibromoanthraquinone (549 mg, 1.498 mmol) under the same method as LNU-26.

### 3.4. Preparation of the Superhydrophobic Fabrics

Using LNU-26 as an object, a piece of polyester fabric (40 × 40 mm) was ultrasonically cleaned in ethanol, deionized water, and acetone (40 mL) for 30 min to remove any stains and oils. After that, the fabric was dried in an oven at 60–70 °C. An amount of 30 mg of LNU-26 powder was dispersed into 20 mL of tetrahydrofuran; after being sonicated for 1 h, the solution was dipped on the polyester fabric to obtain a superhydrophobic flexible fabric coated with LNU-26 powder.

### 3.5. Filtering Experiment

The fabric was first placed in the middle of the filtering apparatus, and then the oil/water mixture was poured from the top. The separation performance of oil (dyed with methyl red) and water (dyed with methyl blue) was recorded using a digital camera. The oil/water separation efficiency (*r*) of the PAF sample was calculated according to the following equation:(1)$$r\left(\%\right)=\frac{m_{1}}{m_{0}}\times 100\%$$
where *m*_0_ is the initial oil weight (g), and *m*_1_ is the weight of oil collected from the oil/water mixture.

### 3.6. Preparation of the Self-Cleaning Glass Sheet

A small piece of double-sided tape was stuck on the glass piece. After that, the PAF powder was directly adhered to the surface of the glass piece by using double-sided tape.

## 4. Conclusions

We demonstrated the synthesis of two different polarities of superhydrophobic porous aromatic skeletons with respective orthoposition and paraposition C=O groups in the PAF linkers. The resulting PAF solids showed high thermal stability and excellent superhydrophobicity. Through a dip-coating process, the PAF-powder-coated fabrics achieved outstanding oil/water separation efficiency of over 90%. Polar LNU-26 PAF showed better separation performance for the polar oils. This work provides powerful theoretical guidance for the industrialization of actual sewage treatment.

## Data Availability

All data related to this study are presented in this publication.

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
