# Peer review of "Introducing Polar Groups in Porous Aromatic Framework for Achieving High Capacity of Organic Molecules and Enhanced Self-Cleaning Applications"

_molecules, 2022, doi:10.3390/molecules27186113_

Round 1
Reviewer 1 Report
In this work, two porous aromatic frameworks (PAFs) were synthesized with ortho- and para- position C=O groups in the PAFs linkers. Then, the PAF powders were coated on polyester fabrics by a simple dip-coating method. Results showed that the oil/water separation efficiency of the flexible fabrics is more than 90% for various organic solvents. This manuscript is well written. The topic of oil/water separation is of significance. I recommend this work be published after minor revision. Please refer to the following suggestions. Please note that these are suggestions only, not all supplementary experiments are necessary.
1.Lines 33-34, authors described that ”With the gradual improvement of people’s living standards, frequent organic waste and oil leakage has caused serious panic [1-4]. ” The relationship between improvement of people’s living standards and leakage might be uncertain.
2. The authors claimed that “This work takes the lead in adopting the polar groups as building units for the preparation of porous networks”. It is better to compare the oil/water separation performance with the PAFs without the polar groups. It lacks the controlled experiments.
3.Line 187, in the Preparation of the superhydrophobic fabrics, how about the firmness (abrasion resistance) of the superhydrophobic fabrics by the dip-coating method.
3.Can the PAFs be recycled after oil/water separation?
4.How about the cost of the PAFs, it matters if they were used in the real case.
5. The superhydrophobic fabrics can be safely degraded or incinerated if need after use?
Author Response
Point-to-point response to reviewer’s comments
Reviewer 1
Question 1: Lines 33-34, authors described that “With the gradual improvement of people’s living standards, frequent organic waste and oil leakage has caused serious panic [1-4].” The relationship between improvement of people’s living standards and leakage might be uncertain.
Response 1: Thanks for your suggestion, which helps us a lot to improve the quality of the manuscript. According to the suggestion, we have made change to the manuscript.
“With the rapid development of industry and transport, frequent organic waste and oil leakage has caused serious panic. (Please see Line 34, page 1 in the revised manuscript)”
Question 2: The authors claimed that “This work takes the lead in adopting the polar groups as building units for the preparation of porous networks”. It is better to compare the oil/water separation performance with the PAFs without the polar groups. It lacks the controlled experiments.
Response 2: Many thanks for your comments. In the manuscript, the polar LNU-26 is composed of 2,7-dibromo-9,10-phenanthrenedione containing an asymmetric ortho-carbonyl group. For comparison, we prepared non-polar LNU-27 containing symmetrical para-carbonyl functional group 2,6-dibromoanthraquinone. It can be seen from Figure 4 that the performance comparison of polar LNU-26 and non-polar LNU-27 in oil-water separation. Polar PAF LNU-26 shows better separation performance for the polar oils.
Question 3: Line 187, in the Preparation of the superhydrophobic fabrics, how about the firmness (abrasion resistance) of the superhydrophobic fabrics by the dip-coating method.
Response 3: Thanks for your comment. We did test the stability of PAF attached to the fabric (Adv. Mater. Interfaces 2022, 9, 2101994). The fabric is rolled up and folded in half more than 100 times to test the stability of the powder attached fabric. The resulting fabric keep the superhydrophobic quality as it is newly prepared.
Question 4: Can the PAFs be recycled after oil/water separation?
Response 4: Thanks for your question. We tested the cycle performance of PAF, and the separation efficiency was still over 93% after 10 cycle experiments (Adv. Mater. Interfaces 2022, 9, 2101994).
Question 5: How about the cost of the PAFs, it matters if they were used in the real case.
Response 5: Thanks for your question. The cost of the PAFs is ca. 50 dollar per gram. Although the price is slightly higher, the excellent performance of PAF materials can make up for the disadvantage of price.
Question 6: The superhydrophobic fabrics can be safely degraded or incinerated if need after use.
Response 6: Thanks for your question. The polyester fabric will decompose completely at the temperature around 450 ° (Ind. Eng. Chem. Res. 2014, 53, 3917, J. Appl. Polym. Sci., 106, 3521). As for PAF materials, they will decompose by more than 90% above 600 ° (Adv. Mater. Interfaces 2022, 9, 2101994, J. Colloid Interface Sci. 2022, 628, 1023-1032). In summary, PAF-coated fabric can be degraded or incinerated after reaching a certain temperature.

Reviewer 2 Report
The authors reported an experimental work of synthesizing two superhydrophobic PAF solid powders (named LNU-26 and LNU-27) with respective ortho-position and para-position C=O groups. The obtained powders were coated on polyester fabrics to prepare two superhydrophobic fabrics. The results have shown that the separation efficiency of the prepared materials for aromatic organic molecules was highly received for potential applications such as oil/water separation or pollutant removal. This work pays more attention toward the environment and contamination related sewage treatment and water purification process.
Overall, I have found the paper well written and introduced an additional understanding of superhydrophobicity and self-cleaning applications.
Author Response
Point-to-point response to reviewer’s comments
Reviewer 2
Comment: The authors reported an experimental work of synthesizing two superhydrophobic PAF solid powders (named LNU-26 and LNU-27) with respective ortho-position and para-position C=O groups. The obtained powders were coated on polyester fabrics to prepare two superhydrophobic fabrics. The results have shown that the separation efficiency of the prepared materials for aromatic organic molecules was highly received for potential applications such as oil/water separation or pollutant removal. This work pays more attention toward the environment and contamination related sewage treatment and water purification process.
Overall, I have found the paper well written and introduced an additional understanding of superhydrophobicity and self-cleaning applications.
Response: Thanks for your comment, your comment is of great help in improving the quality of the article.
